# The Effects of Crosslinking on the Rheology and Cellular Behavior of Polymer-Based 3D-Multilayered Scaffolds for Restoring Articular Cartilage

**DOI:** 10.3390/polym13060907

**Published:** 2021-03-16

**Authors:** Yaima Campos, Francisco J. Sola, Gastón Fuentes, Luis Quintanilla, Amisel Almirall, Luis J. Cruz, José C. Rodríguez-Cabello, Yasuhiko Tabata

**Affiliations:** 1Centro de Biomateriales, Universidad de La Habana, ave Universidad e/G y Ronda, Vedado, Plaza, La Habana CP 10400, Cuba; y.campos.mora@gmail.com (Y.C.); francisco.sola@biomat.uh.cu (F.J.S.); amisel@biomat.uh.cu (A.A.); 2TNI Group, Department of Radiology, LUMC, Albinusdreef 2, 2333 ZA Leiden, The Netherlands; l.j.cruz_ricondo@lumc.nl; 3Laboratory of Biomaterials, Department of Regeneration Science and Engineering, Institute for Frontier Life and Medical Sciences, Kyoto University, 53 Kawara-cho Shogoin, Sakyo-ku, Kyoto 606-8507, Japan; yasuhiko@infront.kyoto-u.ac.jp; 4Bioforge Group, Campus Miguel Delibes, CIBER-BBN, Universidad de Valladolid, Edificio LUCIA, Paseo Belén 19, 47011 Valladolid, Spain; luisq@ele.uva.es (L.Q.); roca@bioforge.uva.es (J.C.R.-C.)

**Keywords:** multilayer scaffold, polymer-based biomaterials, rheology, cellular behavior

## Abstract

Polymer-based tri-layered (bone, intermediate and top layers) scaffolds used for the restoration of articular cartilage were prepared and characterized in this study to emulate the concentration gradient of cartilage. The scaffolds were physically or chemically crosslinked. In order to obtain adequate scaffolds for the intended application, the impact of the type of calcium phosphate used in the bone layer, the polymer used in the intermediate layer and the interlayer crosslinking process were analyzed. The correlation among SEM micrographs, physical-chemical characterization, swelling behavior, rheological measurements and cell studies were examined. Storage moduli at 1 Hz were 0.3–1.7 kPa for physically crosslinked scaffolds, and 4–5 kPa (EDC/NHS system) and 15–20 kPa (glutaraldehyde) for chemically crosslinked scaffolds. Intrinsic viscoelasticity and poroelasticity were considered in discussing the physical mechanism dominating in different time/frequency scales. Cell evaluation showed that all samples are available as alternatives to repair and/or substitute cartilage in articular osteoarthritis.

## 1. Introduction

Disorders of articular cartilage represent some of the most common and debilitating diseases encountered in orthopedic practice. Understanding the normal functioning of articular cartilage is a prerequisite to understanding its pathologic processes. The mechanical properties of articular cartilage arise from the complex structure and interactions of its biochemical constituents: mostly water, electrolytes, and a solid matrix composed primarily of collagen (COL) and proteoglycan. Cartilage’s viscoelastic properties, due to or caused by fluid flow through the solid matrix, can explain many of the deformation responses observed under many loading conditions [1].

Articular cartilage is a highly complex tissue subjected to severe mechanical stress and with very limited regenerative ability. In contrast to elastic cartilage, it is exposed to recurring partial dehydration owing to ongoing compression, but maintains its functionality over decades [2,3]. From the tribological standpoint, articular cartilage is a highly efficient rubbery surface, because it has a low wear rate and a low friction coefficient. Many osteoarticular diseases affect articular cartilage tissue, thereby damaging the cartilage and leading to pain and articular dysfunctions [4].

For the last two decades, the concept of tissue engineering has promised healing of damaged tissues and organs, using living and functional constructs. By manipulating cells, scaffolds, and several kinds of stimuli, the premise was that implanted materials could be generated and subsequently integrated into native tissues and restore lost functions due to trauma, disease, or aging [2,5]. The primary goal of all approaches in tissue engineering is the restoration of function through the delivery of living elements which become integrated into the patient. Although some techniques of guided tissue regeneration are based only in matrices, and other approaches only in cells, most researchers in tissue engineering use a synergistic concept: cells combined with matrices to achieve better performance in the new tissue formation [5].

Mechanical properties are essential for the biological functions and mechanical behavior of the cartilage. These characteristics are primarily dependent on the hierarchical organization in the two macromolecular major extracellular matrix (ECM) components of the cartilage: the fibrillar collagen network and the glycosaminoglycan (GAG)-substituted proteoglycan, mainly aggrecan, aggregates. The delicate balance between swelling pressure and restraining stress controls the mechanical properties of cartilage, which in turn, are essential for articulation, loading, energy dissipation, and diffusion of solutes. Dissipation of mechanical energy in articular cartilage subjected to dynamic loads during walking is essential for protecting bone structures from mechanical stress in joints [1,6].

Platelet rich plasma (PRP) has been used as a natural bioactive scaffold in dentistry for a few years. When it is autologous, it is the quintessential three-dimensional scaffold due to its natural biocompatibility and biodegradability. This is mainly due to the fact that PRP contains several growth factors, such as transformative, endothelial, fibroblast, and platelet-derived. PRP combined with stem cells has shown the ability to regenerate soft and hard tissues [7,8,9].

In normal adult cartilage, collagen is woven together to form a fibrous network in which the huge proteoglycan aggregates are trapped. Together, they form a cohesive porous composite organic solid matrix. The collagen network is characterized by its great tensile stiffness, but due to the high slenderness ratio of each segment of the collagen fibril, the network is relatively weak in compression [10]. In studying the deformational characteristics of articular cartilage under mechanical loading, one of the central concerns has been the determination of the “elastic modulus” of this thin layer of tissue at the ends of the bone in a diarthrodial joint. Due to its anatomical form and its thinness, indentation experiments were the historical choice. The assumption that cartilage is purely elastic applies, at best, only at equilibrium when there would be no dissipative effect due to the movement of the interstitial fluid. It was not until the early 1960s that the systematic study of the role of fluid flow in the function of cartilage actually began [1,10,11].

It is apparent from experimental results, such as those obtained from indentation tests, that the movement of the interstitial fluid plays a fundamental role in the dynamic deformational behavior of the tissue. In other words, to obtain a realistic rheological model for biomechanical studies on articular cartilage, it is necessary to assume the fluid component as a distinct phase of the system within the tissue. This means that, at the very least, cartilage should be modelled as a biphasic material, where the solid matrix and the interstitial fluid are the two phases [1,10,12,13].

Since novel biomaterials to be used in biomedical applications should mimic the natural organs and tissues, a detailed characterization of the former is essential. Natural and synthetic polymers provide an excellent platform for tissue engineering and regenerative medicine strategies. Examples of natural polymers are chitosan (CHI), hyaluronic acid, sodium alginate, and cellulose [14,15]. Among the synthetic polymers, there is broad representation from the vinyl family (acrylic and methacrylic acid; hydroxyethyl and methyl methacrylate), the poly-LG family (poly-lactic acid, poly-glycolic acid and their combination) [16,17], and finally, those that mimic natural structures, such as elastin-like recombinants (ELRs) [18,19,20,21].

Among the protein-based polymers, elastin-like recombinant polymers (ELRs) are emerging as a new class of polymers with exceptional properties. These include mechanical properties ranging from excellent elasticity to plasticity, outstanding biocompatibility, and acute smart behavior. This last attribute is caused by the so-called “inverse temperature transition” (ITT). The ITT has become the key issue in the development of new peptide-based polymers as molecular machines and materials. The understanding of the macroscopic properties of these materials in terms of the molecular processes taking place around the ITT has established the basis for their functional and rational design [22,23].

All the functional elastin-like polymers exhibit phase transitional behavior associated with the ITT. In an aqueous solution, below a certain critical temperature (T_t_), the free polymer chains remain disordered, random coils in solution that are fully hydrated, mainly by hydrophobic hydration. On the contrary and because of the ITT, above T_t_, the chain hydrophobically folds and assembles to form a phase-separated state in which the polymer chains adopt a dynamic, regular, non-random structure, called a β-spiral, involving one type II β-turn per pentamer, which is stabilized by intraspiral interturn and interspiral hydrophobic contacts. During the initial stages of polymer dehydration, the hydrophobic associations of β-spirals lead to a fibrillar form that grows to a several hundred-nanometer particle before settling into the visible phase-separated state [18,19,22,23,24].

Normally, ionic crosslinking in this type of materials is related to the abundance of electrical charges due to the solubility of the polymers used, or at least their interactions with the liquids with which they are mixed or suspended (through electrostactic interactions or hydrogen bonds). For example, this great difficulty for the production of bioinks for 3D printing has been widely studied in the last decade, but its solutions are usually very technologically complicated [25,26] or generate possible collateral products that disadvantage the material balance [27].

The imine formation mechanism is not so simple as to pretend that a simple mixture of cationic (amine type) and anionic (carboxylate type) polymers will spontaneously generate an imine structure. This process requires the use of catalysts and very specific reaction conditions to obtain it [28].

Perhaps this has been one of the main causes of the application of chemical crosslinkers in this type of process, which is also conditioned by the need to increase the mechanical properties of the material and in accordance with its possible field of application. Although there are many crosslinking methods, photosensitive ones with the use of pluronic-type poloxamers [29] and chemical procedures that usually use substances such as gelatin/methacryloil [30], glutaraldehyde [31], and *N*,*N*-(3-dimethylaminopropyl)-*N*′-ethyl carbodiimide (EDC) [32,33] stand out for their abundance today. All these methods have advantages and disadvantages related to their possible applications, and it is a researcher’s task to make the most appropriate selection of each of them.

Normally the cell studies use a primary biocompatibility index, either for adhesion or for cell proliferation. This fact has importance due to lack of cells and nutrients of such a structure in the specific case of the materials for cartilage substitution or regeneration. Most of these works center on chondrocyte studies [34,35], or on other types of cells, such as human umbilical veins, endothelial cells [16], bone marrow mesenchymal stem cells [36], MC3T3-E1 osteoblasts from mice [37], and many others. However, they have extensively reported the necessity of growth factors to improve the proliferation of cells inside of the scaffolds [19,34,37].

This study focuses on the preparation of tri-layered scaffolds with concentration gradients and potential applications in cartilage tissue engineering. For that, we have developed a collagen/chitosan-based multi-layered scaffold with distinct but seamlessly integrated layers that mimics the structure and composition of osteochondral tissue. The interlayer crosslinking is based on either physical (electrostatic type) or chemical interactions (due to amide formation through chemical crosslinking). It could be hypothesized that an ideal scaffold for osteochondral repair could be produced by combining a base layer consisting of a collagen/chitosan scaffold exhibiting osteoinductive properties and potential for bone repair (due to the presence of calcium phosphate materials) with a polymer-based intermediate cartilage layer, and finally, the tri-layered scaffold would be completed with a collagen/chitosan-based cartilaginous layer with a different ratio than the intermediate layer.

The influences of (i) the change of the type of calcium phosphate in the bone layer, (ii) the substitution of the polymer in the middle layer by ELR, and (iii) the interlayer’s crosslinking effect on the scaffold’s properties were evaluated. The mechanical behavior of scaffolds is discussed on the basis of the biphasic porous-viscoelastic (BPVE) model [2,3,4,18,19,38,39] and the contributions of both the fluid-dependent (poroelasticity) and fluid-independent (intrinsic viscoelasticity) elasticityhave been taken into account in this work.

## 2. Materials and Methods

### 2.1. Materials

All chemicals were of analytical grade. They were purchased from Sigma-Aldrich Co. (Madrid, Spain and Tokyo, Japan) and used as received. Another materials’ origins or specifications will be pointed out when necessary. The infrared analysis was done with a Bruker T27 + Opus Data Collection Program (Bruker, Billerica, MA, USA) and the XRD patterns were obtained with a Philips PW1800 Powder X-Ray Diffractometer (Philips Health Systems, Amsterdam, The Netherlands) with Q–Q–Bragg–Brentano geometry, anode Cu (Kα), running at 40 kV and 35 mA.

#### 2.1.1. Ca-P Materials

Octacalcium phosphate (OCP). Ca_8_(HPO_4_)_2_(PO_4_)_4_·5H_2_O, was obtained by dropwise addition of a calcium acetate solution (250 mL, 0.04 mol/L) into a sodium acid phosphate solution (250 mL, 0.04 mol/L), 5 < pH < 6, maintained at 60 °C for 4 h. The solution was unstirred during all the precipitation process and the subsequent digestion periods. Finally, the precipitates were filtered, washed several times with bi-distilled water, and air-dried at room temperature [40]. FTIR (cm^−1^) (Appendix A) [40,41,42]: 1019, 1105 $\left\{\nu_{3}\left(PO_{4}^{3-}\right)\right\}$; 962 $\left\{\nu_{1}\left(PO_{4}^{3-}\right)\right\}$; 561, 606 $\left\{\nu_{4}\left(PO_{4}^{3-}\right)\right\}$; 855 $\left\{\nu_{2}\left(CO_{3}^{2-}\right)+\nu \left(P-OH\right)\right\}$. DRX (2θ/°) (Appendix A) (ASTM 26-1056) [40,42,43]: 4.74° (0 1 0), 9.45° (0 2 0), 9.81° (1 1 0), 26.00° (0 0 2), 31.56° (2 6 0), and 31.70° (2 −4 1).

Hydroxyapatite (HAP). Ca_10_(PO_4_)_6_(OH)_2_, was obtained by dropwise addition of a phosphoric acid solution (250 mL, 0.4 mol/L) into a suspension of calcium oxide in bi-distilled water (250 mL, ≈ 0.5 mol/L) under stirring. The end point of the reaction was established by pH at 7.2 ± 0.1. Then, the precipitate was left overnight in the reaction flask without stirring and washed several times with bi-distilled water, dried in a stove at 105 °C for 24 h, heated at 1000 °C by 3 h, and finally, milled and classified below 160 µm [42,44]. FTIR (cm^−1^) (Appendix A) [41,44,45]: 3571 $\left\{\nu \;\left(OH^{-}\right)\right\}$; 1634 $\left\{\delta \;\left(OH^{-}\right)\right\}$; 1037, 1100 $\left\{\nu_{3}\left(PO_{4}^{3-}\right)\right\}$; 962 $\left\{\nu_{1}\left(PO_{4}^{3-}\right)\right\}$; 877 $\left\{\;\nu_{2}\left(CO_{3}^{2-}\right)+\nu \left(P-OH\right)\;\right\}$; 560, 607 $\left\{\nu_{4}\left(PO_{4}^{3-}\right)\right\}$; 606 $\left\{\gamma \;\left(OH^{-}\right)\right\}$. DRX (2θ/°) (Appendix A) (ASTM # 09-0432) [42,44,46,47]: 25.81° (0 0 2), 31.74° (2 1 1), 32.13° (1 1 2), 32.91° (3 0 0), and 49.39° (2 1 3).

#### 2.1.2. Polymeric Materials

COLLAGEN: It constitutes approximately 10% of the wet and 50% of the dry weight of cartilage. Each molecule consists of three polypeptide chains coiled into a unique type of rigid helical structure. The unique properties of the polypeptide chains are in part due to the rigid amino acid repeating triplets of gly–pro–hydroxyproline, although they could change according to collagen type [48,49]. FTIR (cm^−1^) (Appendix A: 3296) (amide A); 3084 (amide B); 1635 (amide I); 1545 (amide II); 1236 (amide III)

CHITOSAN: It is the only pseudo-natural cationic polymer, and is called so when chitin reaches the degree of deacetylation of about 50%. CHI’s solubilization occurs by protonation of the –NH_2_ functional group on the C-2 position of the D-glucosamine repeating unit (the other is N-acetyl-β-D-glucosamine), whereby the polysaccharide is converted to a polyelectrolyte in acidic media [50,51]. FTIR (cm^−1^) (Appendix A): 3326 $\left\{\nu \;\left(OH^{-}\right)+\nu \;\left(NH_{2}\right)\right\};$ 2933 $\left\{\nu \;\left(CH_{3}\right)\right\};$ 2867 $\left\{\nu \;\left(CH_{2}\right)\right\};$ 1563 $\left\{\delta \;\left(NH_{2}\right)\right\};$ 1414 $\left\{\nu \;\left(OH,\;alcohol\right)\right\}$.

ELASTIN: The ELR used in this work, denominated (GEG)_15_, was supplied by the Bioforge Group (University of Valladolid, Spain). Its amino acid sequence is [(VPGVG)_2_VPGEG(VPGVG)_2_]_15_ and it was obtained by standard genetic engineering techniques [22]. Specifically, it was produced using cellular systems for genetically engineered protein biosynthesis in *Escherichia coli* and purified using several cycles of temperature-dependent reversible precipitations, as described in the literature [20,21,22,23,24]. Due to the COOˉ groups of glutamic acid, this ELR increases the negative charge density in the intermediate scaffold layer to favor the simulation of the concentration gradient in the cartilage normally produced by the high negative charge density provided by the proteoglycans [52]. Moreover, through this ELR sequence being incorporated into the material cell-adhesion of RGD and REDV peptide sequences [18], the biological performance of the scaffold is improved. FTIR (cm^−1^) (Appendix A): 3291 (Amide A); 3069 (amide B); 1625 (amide I); 1524 (amide II) and 1236 (amide III).

It can be clearly inferred that the natural polymers used in this work provide amino, carboxylate, and hydroxyl terminal groups according to the most abundant compositions thereof.

### 2.2. Polymeric Solutions

COL (type I collagen fibers from the bovine Achilles tendon, commercial grade, Brazil), CHI, and ELR 2% (*w*/*v*) solutions were left overnight under magnetic stirring in separate flasks; 10 µL of glacial acetic acid (per mL of polymer solutions) was added to each one after 24 h, and stirring continued for another day. On the third day, the three layers’ composites were prepared as described below.

### 2.3. Scaffold Fabrication

The three-layer scaffolds were manufactured with gradient of concentration from the bone layer (or bottom layer that included calcium phosphate materials to improve bone adhesion) up to the cartilage layer (or top layer) going through the intermediate layer. The two upper layers were composed only of natural and synthetic polymers. The preparation of each layer’s suspensions can be seen below (Table 1).

#### 2.3.1. Bone Layer Suspension (B-Layer)

The same volumes (1:1) of 2% collagen solution and 2% chitosan solution, 10 µL of Tween 80 (per mL of layer suspension), and the powder necessary to reach 2% (*w*/*v*) of calcium phosphate in the final volume, either OCP and HAP (**O** and **H** in the sample code of Table 1, respectively) were added into a 100 mL beaker with mechanical stirring at 5000 rpm for 30 min. Then, 16 µL of NaOH 1 mol/L (per mL of layer suspension) was added to neutralize the acetic acid. The mixture was added to the mold and frozen for 30 min at −20 °C (Appendix A).

#### 2.3.2. Intermediate Layer Suspension (M-Layer)

The same volumes (1:1) of 2% collagen solution and 2% chitosan solution (or (GEG)_15_, **E** for ELR in the sample code of Table 1) and 10 µL of Tween 80 (per mL of layer suspension) in the final volume were added into a 100 mL beaker with mechanical stirring at 5000 rpm for 30 min. Then, 16 µL of NaOH 1 mol/L (per mL of layer suspension) was added to neutralize the acetic acid. The mixture was added to the mold and frozen for 30 min at −20 °C (Appendix A).

#### 2.3.3. Cartilage Layer Suspension (T-Layer)

Volumes (3:1) of 2% collagen solution and 2% chitosan solution and 10 µL of Tween 80 (per mL of layer suspension) in the final volume were added into a 100 mL beaker with mechanical stirring at 5000 rpm for 30 min. Then, 16 µL of NaOH 1 mol/L (per mL of layer suspension) was added to neutralize the acetic acid. The mixture was added to the mold and frozen for 30 min at −20 °C (Appendix A).

#### 2.3.4. Final Step

The scaffold was kept for 24 h at −80 °C and lyophilized. Finally, it was gently washed with MilliQ water several times to eliminate all the basic and acidic residues. Then, it was lyophilized again (Appendix A).

### 2.4. Crosslinking Process

During the scaffold manufacturing process, physical crosslinking resulted mainly due to the electrostatic interactions among the amine (COL; CHI; ELR), acid (COL; ELR), and hydroxyl (COL; CHI) groups of the polymers involved in each layer; due to the presence of water, these interactions may be stabilized through hydrogen bonds. In particular, the presence of the glutamic acid in the ELR and its subsequent increase of the negative charge density (COO−) will affect the physical crosslinking in the corresponding scaffold. As an alternative to these relatively weak interactions, chemical crosslinking has been also included to improve the performance of the scaffolds [17,53,54,55].

#### 2.4.1. Crosslinking with Glutaraldehyde

The reaction between carbonyl groups in the glutaraldehyde and amino groups (mainly available in the natural polymers and in minor extensions in the ELR due to its limited amount of terminal amino groups) of the scaffold layers can be described according to the chemical reaction in the Figure 1a (imine formation). No chemical crosslinking is produced between the G carbonyl groups and the ELR carboxyl groups in this case. For chemical crosslinking, the scaffold was immersed in a 2.5% (*v*/*v*) glutaraldehyde solution (**G** in the sample code) for 45 min, and then was wash repeatedly with MilliQ water, frozen at −20 °C, and lyophilized.

#### 2.4.2. Crosslinking with EDC/NHS

The reaction between amine groups in the corresponding (EDC)/N-hydroxy succinimide (NHS) solution and the carbonyl groups of the available in the polymers of the scaffold layers can be described by several steps. The EDC reacts with carbonyl groups and forms an unstable intermediate which should react immediately with three possible compounds (Figure 1b):The NH_2_-terminal provided for natural polymers in the multilayer scaffold resulting, a stable amide bond;The NHS, which is a result of a semi-stable amine NHS ester that forms with the terminal NH_2_ provided by natural polymers a stable amide bond;A small quantity of carbonyl groups are regenerated because the unstable intermediate reacts with water of the medium [32,33].

For chemical crosslinking, the samples were immersed in a solution of EDC/NHS (33 mM/6 mM) in ethanol (95%) for 4 h at room temperature. Then the crosslinking scaffolds were washed (5 times × 5 min), frozen at −80 °C, and lyophilized.

### 2.5. Morphological, Chemical, Physical, and Mechanical Characterization

#### 2.5.1. Microstructural Morphology

Scanning electron microscopy (SEM) was used to investigate the scaffold morphology. Thus, the samples were immersed in MilliQ water at 37 °C for 15 min (maximum swelling); then immediately dropped into liquid nitrogen, physically fractured, and immersed in liquid nitrogen again. Finally, they were freeze-dried. Images of lyophilized scaffolds were obtained by SEM (FEI Quanta 200FEG, Hillsboro, OR, USA) with Schottky’s Filament Field Emission Cannon and voltages of 0.2–30 kV. Micrographs were achieved under ESEM mode at 10 kV and a EDAX Genesis micro-probe (Mahwah, NJ, USA) was used for elemental microanalysis of the scaffold’s different layers.

#### 2.5.2. Swelling Studies

Swelling measurements were performed by a gravimetric method to determine the water percentage by weight absorbed by the scaffolds. Small octahedral pieces of ≈ 30 mg were cut transversally (to include the three layers). The small octahedrons were weighed (*W_i_*) and submerged in saline buffered solution (Dulbecco’s PBS, Nissui Pharmaceutical Co. Ltd., Tokyo, Japan) at pH = 7.4 and 37 °C (Digitheat-TFT, J.P. Selecta, Spain) and weighed again at predetermined time intervals. The difference between the weight of the scaffold at time *t*, *W_t_*, and the initial weight, *W_i_*, defines the water percentage by weight, expressed as swelling degree, *W* (Equation (1)):(1)$$W=\frac{W_{t}-W_{i}}{W_{i}}\times 100=100\frac{W_{t}}{W_{i}}-100$$

The swelling degree tests were carried out until the values remained constant for three consecutive measurements.

#### 2.5.3. Mechanical Analysis

Rheological experiments were performed using a strain-controlled AR-2000ex rheometer (TA Instruments, Newcastle, DE, USA) with the sample submerged in water at 37 °C after 15 min for complete swelling. Cylindrical swollen scaffolds were placed between parallel plates of non-porous stainless steel (diameter = 12 mm). A normal force adequate to prevent slippage was applied. A gap larger than 1000 µm was always reached after the sample relaxed until equilibrium.

Several oscillatory measurements were performed in a shear deformation mode. Initially, the range of strain amplitudes over which the samples exhibited a linear region of viscoelasticity was determined. Thus, a dynamic strain sweep (with amplitudes ranging between 0.01 and 10%) was carried out at a frequency of 1 Hz to measure the dynamic shear modulus as a function of strain. Secondly, dynamic frequency sweep tests were performed to determine the dependence of the dynamic shear modulus and loss factor on frequency. Specifically, a frequency sweep between 0.01 and 10 Hz at a fixed strain (corresponding to the scaffold linear region) was selected.

Finally, to determine the transient evolution of the relaxation modulus, a stress relaxation test was undertaken. One of the potential biomedical applications supports this type of measurement. The scaffolds developed to repair joint cartilage are more effective if their stress relaxation behavior matches that of the native tissue, since such behavior affects load transfer and nutrient transport [10,18].

Rheological measurements provided the storage modulus, the so-called elastic modulus (*G’*), and the loss modulus, the so-called viscous modulus (*G″*), as a function of strain or frequency at 37 °C. The complex modulus magnitude, the so-called dynamic shear modulus (|*G**|^2^ = (*G’*)^2^ + (*G″*)^2^), and the loss factor (tan *δ* = *G″*/*G’*, where *δ* is the phase angle between the applied stimulus and the corresponding response) were also obtained. Moreover, the transient evolution of the relaxation modulus, *G(t)*, was recorded [56].

### 2.6. Cell Studies

Viability assay. The suitability of the scaffolds for the culture of immortalized human chondrocytes (C-28 cell line) [57] was studied. Briefly, 3 × 10^4^ C-28 human chondrocytes per well were seeded on the top of one scaffold sample (3 mm width and length × 1 mm height) with DMEM in a 48-well plate. The scaffold samples were removed after 3, 7, 10, or 14 days of seeding, and then assessed using a Calcein-AM/ethidium homodimer-1 (EthD-1) LIVE/DEAD^®^ assay kit, according to the manufacturer’s instructions. Specifically, the isolated pre-loaded samples were washed with PBS, treated with the assay kit, kept in darkness for 30 min, and finally, studied by confocal microscopy. In the LIVE/DEAD^®^ assay, living cells are stained green, whereas dead cells are stained red [58]. The cells nuclei were stained with DAPI at the same time to corroborate the results of the cells population [59,60]. The materials, the positive and negative controls, have been analyzed by the Leica DM5500 B fluorescence microscope (filter settings: FITC and DAPI), equipped with a Leica DFC365 FX digital camera. Digital images were acquired and stored using Leica Application Suite X (LAS X) software (Leica Microsystems, Amsterdam, The Netherlands).

MTS assay. To further corroborate the cell viability results, an MTS assay was performed. This is a colorimetric technique in which (3-(4,5-dimethylthiazol-2-yl)-5-(3-carboxymethoxyphenyl)-2-(4-sulfophenyl)-2H-tetrazolium), in the presence of phenazine methosulfate (PMS), produces a formazan product that has an absorbance maximum at 490 nm in PBS. Scaffold samples with dimensions like the viability assay were loaded with C-28 human chondrocytes (density: 1 × 10^4^ per well; 500 µL of cell suspension), and then incubated for 3, 7, 10, or 14 days; 100 µL of the supernatant solution was extracted to a 96-well plate for reading at λ = 490 nm into a tunable, spectrophotometric microplate reader (VersaMax, York County, PA, USA with Program Softmax Pro).

### 2.7. Statistical Analysis

Graphs were made and statistics gathered with OriginPro 2018 (OriginLab Corp., Northampton, MA, USA). Data are reported as mean ± standard deviation (SD), unless stated otherwise. Error bars represent the SDs calculated from tests of triplicate measurements for each scaffold. Statistical analysis was significant by a one-way analysis of variance (one-way ANOVA) for *p* < 0.05 (*) or *p* < 0.01 (**).

## 3. Results and Discussion

### 3.1. Physical-Chemical Characterization

#### 3.1.1. Morphological Characterization

Despite the apparently heterogeneous pore structure of the three-layer 3CCO scaffold shown in Figure 2A, a good degree of pore interconnectivity throughout the construct could be observed. Each layer had approximately 2 mm of height with good integration between individual layers, as shown by the lack of visible interfaces among them. The height of each layer can be adjusted by changing the volumes used during the layers’ preparation. This good connection between layers is vital in order to promote cell infiltration and the regeneration of tissue in the different layers of the scaffold, and to obtain adequate mechanical properties.

Scaffold porosity was high for each of the three layers. A slight reduction in porosity was seen in the B-layer containing OCP. In this layer the pore size revealed a diameter distribution from 100 to 400 µm, ideal for the cell proliferation. There was no significant difference in the pore size of different layers, in agreement with similar works reported in the literature [34,61]. Collagen fibers were integrated into the three-dimensional structure of the scaffold, contributing to the interconnectivity and definition of the pore shape [11,61,62,63].

The difference between the base and the intermediate layer can be clearly seen because the B-layer has small, clearly visible white spots corresponding to the calcium phosphate (OCP), whereas it is more difficult to appreciate these particles in the intermediate layer. Nevertheless, a slight change in the grayish tonality can be appreciated—the M-layer being darker and the T-layer being clearer. No particles were detected under our experimental conditions in the top layer.

High intensity signals in the EDX graph (Figure 2B) were observed in the base layer corresponding to Ca and P atoms due to the OCP used in this layer. The EDX’s graph corresponding to the M-layer indicates that Ca and P signals were still clearly observed. During the manufacturing process of scaffolds, the OCP can be found close to the boundary between the layers, and partial melting of the upper part of the B-layer occurs when the M-layer’s solution is added. Thus, some particles of Ca-P could migrate from one layer to another and be trapped after the freezing process in the M-layer.

In the case of the chemically crosslinked 3CCO.G scaffold (Figure 2C), the apparent boundaries were fully lost, suggesting excellent adhesion between layers. Note the decrease in pore size and number of fibers in the inner fracture zone due to the chemical crosslinking process. This structure was more compact than the physically crosslinked scaffold with the same composition due to chemical bonds. From the EDX signals, high Ca and P intensities can be seen in the M-layer (Figure 2D). In a 45-min immersion in a G solution, simultaneously with the reaction of the amino groups of the polymers with the carbonyl groups of the glutaraldehyde, a partial swell took place which led to the release of some of the OCP particles. Thus, migration and distribution of OCP in M-layer is allowed [23,34,54,64]. In addition, a high C signal was observed due to the use of G.

As far as the crosslinking process is concerned, similar behavior was observed for the samples based on COL/CHI in the M-layer. However, when (GEG)_15_ was included in this layer, substantial changes could be observed in the structural morphology of the scaffold. Appendix A shows a noticeable morphological change in the 3CCHE scaffold (with ELR and without G) with respect to the 3CCO and 3CCO.G samples (Figure 2A,B, respectively), where both a higher pore size and a poorly-organized distribution of fibers were detected.

Micrographs in Appendix A show the boundary between the B and M layers for the scaffolds (GEG)_15_ without and with crosslinker, respectively. Different shapes, distributions, and pore sizes were noticed. The sample 3CCHE.G (Appendix A) showed an interlayer region with the covered pores and the Ca-P particles adsorbed in the scaffold, instead of exposed ones as in the 3CCHE (Appendix A) sample.

#### 3.1.2. Swelling Studies

The scaffold´s ability to swell plays an important role during the in vitro culture studies [54,64,65] and in the mechanical properties [62,66,67,68,69]. The swelling degree of the scaffold was calculated by applying conventional Flory–Huggins equation (Equation (1)). The swell scaffold will present a higher pore size and the presence of fluid among them, which should facilitate the colonization of the matrix by the cells. The swelling profiles of the scaffolds (Figure 3) with different calcium phosphates in the B-layer (3CCO and 3CCH) showed no significant differences either in shape or in values. When (GEG)_15_ was used instead of chitosan in the M-layer (3CCHE), more swelling was observed. In all cases the equilibrium (maximum swelling) was reached within about 15 min, regardless of the sample’s composition.

Non-significant differences could be seen in the swelling profiles when reconfiguring the scaffolds with changes in calcium phosphate in the B-layer due to the similarity in structure of OCP and HAP, consistent with the low solubility of both salts in the aqueous medium [70].

On the contrary, the change in the polymer (ELR instead of CHI, 3CCHE) in the M-layer had a great influence, because of the increment of available COO− groups and their impact on the physical crosslinking. This should have led to drastic changes in the mechanical behavior of material, which will be evaluated in the next subsection. The addition of glutaraldehyde (3CCO.G, 3CCH.G, and 3CCHE.G) and EDC (3CCH.N) as crosslinkers caused a drastic and significant decrease in the swelling capacity of all the scaffolds (Figure 3). On the one hand, this was accomplished by the chemical reactions between the carbonyl groups of glutaraldehyde and the amino groups of the polymers, and on the other hand, by the reactions between the carboxylate groups of the scaffolds and one of the nitrogens of the carbodiimide’s double bond.

The primary factor that affects the swelling ability is the chemical crosslinking procedure. The water-binding ability of the scaffolds could be attributed to both their hydrophilicity and the maintenance of their three-dimensional structure. In general, the swelling ratio decreases as the crosslinking degree increases, because of the diminishing of the hydrophilic groups and the higher stiffness of the scaffold. The absolute value decreases approximately 1.5 times over its initial weight after the chemical crosslinking process, which is high enough for tissue engineering.

These experimental results agree with those reported in the bibliography. The weight gain of poly-ε-caprolactone/elastin composites 2-fold from 15.8 ± 0.3 *w*/*w* to 38.3 ± 0.7 *w*/*w* when the elastin solution concentration was increased up to a saturation level >50 mg/mL [34]. Samples with pure chitosan hydrogel showed the highest water content (28.14 g/g), while in the case of structures prepared from chitosan/glycerophosphate homogenous solutions and two-layer scaffolds, the addition of silk fibers to the hydrogel reduced the water content to 27.02 and 25.5 g/g, respectively. The high water content of scaffolds allows for the transport of nutrients and waste through the matrix [66].

### 3.2. Rheological Properties

The mechanical/viscoelastic properties of these scaffolds must be considered, since they are key properties for the functionality of the scaffold. In this sense, it is well known that rheological measurements are an adequate tool for characterizing these properties [18].

The viscoelastic mechanical properties of the scaffold were determined by rheological measurements over the linear viscoelastic range, providing information under conditions close to the unperturbed material state. Accordingly, in compliance with the principle of small deformation rheology, each sample was tested over its respective linear viscoelastic range [71,72,73]. The evolution of the complex modulus magnitude with the strain amplitude can be found for each sample in Appendix A. All subsequent rheological tests were performed using the same value of strain (0.2%), which was consistent with the linear viscoelastic range for each scaffold.

#### 3.2.1. The Frequency Responses of Moduli

The frequency evolution models of G’ and G″ have been obtained, and both are frequency-dependent, as can be seen in Figure 4. Similar frequency-dependencies have been reported for biological tissues in the literature for human and animal tissues such as liver [74], uterus [75], the adventitial layer of a pig [76], canine kidney cortex and marrow [77], isolated chondrocytes from articular cartilage [78], brain and nerves, liver, fat, relaxed muscle, breast gland tissue, dermis, connective tissue, contracted muscle epidermis, cartilage [56], nucleus pulposus, eye lens [77], and newly synthesized materials [4,18,19,24,56,66,79,80]. The values of both moduli at a frequency of 1 Hz have been summarized in Table 2. For every sample measured, G’ ≫ G″ indicates high elastic behavior. Thus, G’ is the major contribution to |G*|.

In the case of the physically crosslinked scaffolds (Figure 4a), similar storage modulus values in the range 1–2 kPa were obtained for both 3CCO and 3CCH. Yet, when (GEG)_15_ was used in the M-layer, this value decreased almost four times. It could be stated that the drastic decrease in the storage modulus was due to the use of the recombinamer (ELR), whose remarkable increase of COOˉ groups [18,19,23,24] with respect to the chitosan contribution provided to the 3CCHE scaffold more sites of negative charges. The carboxyl groups of the ELR increased the negative charge density in both interlayers (B-M and M-T layers), acting as a barrier that hinders the physical crosslinking through electrostatic forces. In consequence, the values of the mechanical properties were reduced. Nevertheless, the loss modulus showed very similar trends for all the samples with values ranging from 50 to 110 Pa; non-significant differences were found among the three physically crosslinked scaffolds.

In the case of chemically crosslinked scaffolds, G’ and G″ values are at least one order of magnitude higher than the values of the physically crosslinked samples with G (Figure 4b). The fact of the reaction between the glutaraldehyde and the amino groups of the different polymers used in the preparation of the scaffold clearly shows that the interlayer interaction is transformed from being just physical to being mostly chemical, structurally supported by the large number of imine groups formed by chemical crosslinking. No significant differences were found among any of the chemically crosslinked samples.

We should point out the increase of the storage modulus for 3CCHE.G with respect to the sample without G due to the chemical crosslinking (Table 2) [64,81]. It can be suggested that the COL contribution to the crosslinking density dominates, and therefore, it is responsible for most of the mechanical scaffold properties. Some contribution may also be attributed to the higher quantity of amino groups available in the ELR when this polymer is used in the middle layer. The increase of just four times approximately, was obtained in the crosslinking process with EDC, just as it was foreseen from the swelling studies (Appendix A, Figure 4b and Table 2). Nevertheless, normally this EDC/NHS’s system is the most used because for its solubility, the crosslinking’s excess is eliminated easily and with low cytotoxicity [32,33]; in the meantime, the removal process of glutaraldehyde is the most complicated from the technologic point of view [64].

**Table 2 polymers-13-00907-t002:** Superscript capital letters are used to compare between scaffold values (columns) with matching references values of the literature in the same row. Hz.

Magnitude	3CCO	3CCH	3CCHE	3CCO.G	3CCH.G	3CCH.N	3CCHE.G	Approx. Reference Values
G’ (kPa)	1.71 ± 0.02 ^A^	1.4 ± 0.3 ^A^	0.33 ± 0.05 ^B^	19 ± 7 ^C^	15 ± 3 ^C^	3.86 ± 0.07 ^D^	18 ± 8 ^C^	1.8–7.5 ^(AD)^ [18] 1–10 ^(AD)^ [24] 0.01–3.5 ^(ABD)^ [56] 0.006–1.000 ^(B)^ [36] 0.3 ^(B)^ [66] 0.5–2.7 ^(AB)^ [79] 0.1-1 ^(B)^ [82]
G″ (kPa)	0.11 ± 0.01 ^A^	0.07 ± 0.01 ^B^	0.05 ± 0.01 ^C^	1.5 ± 0.2 ^D^	1.4 ± 0.4 ^D^	0.3 ± 0.1 ^E^	1.3 ± 0.6 ^D^	0.02–0.75 ^(ABE)^ [18] 0.02–0.40 ^(ABE)^ [24] 0.001–0.030 [36] 0.25 [66] 0.01–0.04 [82]
tan	0.064 ± 0.008 ^A^	0.05 ± 0.02 ^A^	0.16 ± 0.06 ^B^	0.08 ± 0.04 ^C^	0.09 ± 0.04 ^C^	0.08 ± 0.02 ^C^	0.07 ± 0.07 ^C^	0.19–0.22 [4] 0.07–0.11 ^(AC)^ [18] 0.061–0.087 ^(AC)^ [19]♦ 0.01–0.06 ^(A)^ [24] 0.096–0.19 ^(BC)^ [80]♦ 0.033-0.045 [83]
δ (°)	3.7 ± 0.5 ^A^	3 ± 1 ^A^	9 ± 3 ^B^	5 ± 2 ^AC^	5 ± 2 ^AC^	4.6 ± 0.9A ^C^	4 ± 4 ^AC^	10.7–12.4 ^(B)^ [4]♦ 4.0–6.3 ^(C)^ [18]♦ 3.5–5.0 ^(A)^ [19] 0.5–3.5 ^(C)^ [24]♦ 5.5–11 ^(BC)^ [80] 1.8–2.6 ^(C)^ [83]♦
G* (kPa)	1.7 ^A^	1.4 ^A^	0.33 ^B^	19 ^C^	16 ^C^	3.9 ^D^	18 ^C^	200–250 [4] 1.8–9.5 ^(AD)^ [18] 2.0–5.5 ^(D)^ [19] 1.25–2.00 ^(A)^ [80]
$\tau_{1}$(s)	8 ± 4 ^A^	7 ± 1 ^A^	1.0 ± 0.5 ^C^	8.4 ± 0.2 ^A^	12.3 ± 0.4 ^B^	8 ± 3 ^A^	23 ± 1 ^D^	8–10 ^(AB)^ [18]
$\tau_{2}$(s)	50 ± 10 ^A^	150 ± 30 ^B^	18 ± 5 ^C^	77 ± 2 ^D^	85 ± 2 ^E^	40 ± 10 ^A^	180 ± 30 ^F^	100–110 [18]
$\tau_{3}$(s)	530 ± 60 ^A^	1400 ± 400 ^B^	270 ± 20 ^B^	860 ± 30 ^C^	820 ± 20 ^C^	460 ± 50 ^A^	1200 ± 300 ^D^	1000–1200 ^(BD)^ [18]

^♦^ Values calculated from reported data.

These moduli values agree with those reported for similar materials. Specifically, G’ and G″ values of 300 and 250 Pa, respectively, have been reported for thermosensitive chitosan hydrogels crosslinked with silk fibers [66]. However, if the silk it is substituted by β-glycerophosphate, the moduli decrease to 100 Pa for G’ and 10 Pa for G″ [82]. In both cases, the crosslinking process was physical, but when the chemical crosslinking was included these values oscillated between 1.8–7.5 kPa and 20–750 Pa for G’ and G″, respectively [18]. Meanwhile, other ELRs with different protein sequences but similar chemical crosslinking process had respective values of 1–10 kPa and 20–400 Pa [24]. Huang et al. [36] reported G’ and G″ intervals of 6–1000 Pa and 1–30 Pa, respectively, for materials with the same chemical composition (COL/CHI/HAP).

As far as the loss factor is concerned (Appendix A), a slight frequency dependence was observed for the physically crosslinked scaffolds. The values reported in Table 2 indicate non-significant differences between 3CCO and 3CCH at the frequency of 1 Hz, (around 3-4°). The phase angles are very low, corresponding to highly elastic, energy-storing polymers. The higher values of tan δ for 3CCHE may be related to the presence of ELR in the M-layer, thereby increasing the negative charge density and reducing physical interactions in the interlayer surfaces [18,19,24]. In the case of chemically crosslinked scaffolds, a clear frequency dependence was found, and the loss factor was higher than that of the physically ones, mainly at low frequencies (4–5°). When frequency increases, the difference is reduced and the phase angle at 1 Hz ranges from 3° to 5° (see Table 2) for each sample (excluding 3CCHE), regardless of the crosslinking type or agent.

Thus, although the crosslinking process drastically influences the individual values of the storage and loss modules, the ratio G″/G’ was relatively similar for all the samples at 1 Hz (excluding 3CCHE). Similar values and conclusions of these magnitudes were obtained in previously reported scaffold studies focused on cartilaginous restoration, indicating the clear dominance of the elastic behavior over the loss one, even with the inclusion of Ca-P particles or crosslinking process, as reported in Table 2 [4,18,19,24,80,83].

#### 3.2.2. Poroelasticity and Intrinsic Viscoelasticity

Viscoelasticity is defined as the time-dependent response of a material that has been subjected to a load or deformation. Fluid flow through the cartilage solid matrix is the primary factor responsible for the viscoelastic behaviors observed in articular cartilage. Considering that articular cartilage is mostly loaded under compression, understanding its compressive properties is essential to comprehending this tissue’s function. Under compression, a volumetric variation, and as consequence a pressure change in the tissue occurred. This resulted in the flow of interstitial fluid through the extracellular matrix, generating frictional resistance.

Therefore, the tissue’s behavior under compression is a result of its biphasic nature and the forces which act to balance the externally applied load. This biphasic nature of cartilage could be described by the three major forces which act on the surfaces of the tissues. The forces are caused by: (1) the stress in the solid phase; (2) the pressures in the fluid phase; and (3) the frictional resistance because of the fluid flow in the solid phase [1].

The dependence of |G*|on f^½^ has been plotted in Appendix A for all the scaffolds studied. Two different regions are clearly observable [18]. A non-linear relationship was found at low frequencies (around 0.01–0.25 Hz), corresponding to the region where the intrinsic (fluid-independent) viscoelasticity mechanisms dominate. In contrast, a linear relationship was observed in the ranges 0.25–10 Hz and 0.25–3 Hz for chemically and physically crosslinked scaffolds, respectively. In this frequency range, poroelasticity mechanisms dominate. At higher frequencies, this linear dependence is lost.

The slope of the linear region (Table 3) has been calculated using a least-squares fitting of the experimental data (see Appendix A). Significant differences can be observed between the slope values ($d\left|G^{*}\right|/d\sqrt{f}$) in its linear region.

Considering that G’ is the dominant contributor to |G*|, it may be suggested that the slope change observed corresponds to a permeability change in the scaffolds. It is well known that permeability of a hydrogel is a macroscopic measure of the ease with which a fluid can flow through the matrix. Thus, permeability decreases as the matrix becomes denser and more compact.

The lowest slope value for 3CCHE corresponds to the highest pore size observed in Appendix A. The chemical crosslinking gave rise to a noticeable decrease in the scaffold’s permeability with respect to that of the physically crosslinked scaffold, in agreement with the scaffold morphologies shown in Figure 2 and Appendix A. In the case of 3CCH.N, the explanation of the slope’s low value could be the difference in strength between the bonds (imine vs. amide) exposed above. The slope values obtained in this work are comparable to those reported for materials of similar composition. For instance, elastin-like recombinamer catalyst-free click gels had values in the range of 200–800 Pa/Hz^½^ [18].

#### 3.2.3. Time Relaxation Modulus

In a stress relaxation test, a shear strain is applied to the scaffolds at *t* = 0 s and kept constant up to the end of the measurement while recording the corresponding relaxation modulus, namely, the ratio of the shear stress required to maintain a fixed strain to this same strain, as a function of time. A final time of 1800 s was selected to determine long-term scaffold behavior. Due to the transient response of the rheometer, it showed the evolution of the composite relaxation modulus for *t* > 5 s. As such, no values were obtained for relaxation processes faster than that time.

The flow-independent viscoelastic behavior in cartilage and the viscoelastic nature of collagen fibers and proteoglycan have been demonstrated [3,10,55,84,85]. These observations indicate that the BPVE model—which takes into account the viscoelastic behavior generated from the flow-dependent frictional interactions, and the flow-independent viscoelastic nature of the porous solid matrix—is essential for modeling the behavior of soft tissues and hydrogels [18,85].

When a fluid-independent viscoelastic mechanism (intrinsic viscoelasticity) dominates the relaxation modulus, *G(t)*, the time evolution of this parameter can be described by the equation [18,85,86]:(2)$$G\left(t\right)=G_{eq}+\sum_{i}^{N}G_{i}\exp\left(-t/\tau_{i}\right)$$
based on a series of decreasing exponential functions (Figure 5), where *τ_i_* and *G_i_* are the relaxation time constant and the weight or degree of contribution of the *τ_i_*-type relaxation to the overall relaxation process, respectively; and *G_eq_* is the equilibrium modulus, $G_{eq}=\underset{t\to \infty }{\lim}G\left(t\right)$. For cartilage, the value of *N* = 3 (three different relaxation processes) in the discrete representation has been shown to be sufficient to model the change in the relaxation modulus predicted by the continuous spectrum model, which uses an integral representation [85].

The longer time constants (τ_2_ and τ_3_, see Table 4) were higher for 3CCH than 3CCO, giving rise to a slower transient. Thus, whereas the transient for 3CCO was stabilized at t = 1800 s, the corresponding transient for 3CCH did not reach equilibrium by then. It can be suggested that the presence of HAP hampers and complicates the relaxation processes.

Finally, the chemical crosslinking with glutaraldehyde led to relatively similar transients for the three scaffolds. Nevertheless, the three relaxation processes for 3CCHE.G showed higher time constants than did 3CCO.G and 3CCH.G. It can be suggested that the electrostatic interaction between positive Ca^2+^ ions in the B-layer and negative COOˉ groups of the ELR in the middle layer [17,37] might contribute to these high time constant values.

Transients were measured for a strain amplitude of 0.2% at 37 °C for all the samples. All the samples were fitted to three relaxation process (*N* = 3), with the exception of the 3CCHE experimental data, which were fitted to only two (*N* = 2) to obtain fitting parameters with physical meaning. Once again, the 3CCHE scaffold showed different behavior due to the loss of physical crosslinking—now from the viewpoint of the relaxation modulus. The parameter values obtained by fitting the experimental data to Equation (2) (Appendix A) are reported in Table 4. Each time constant dominated the relaxation process within its corresponding time range. Thus, the transient tail was mainly controlled by the relaxation process characterized by the longest time constant.

At a given temperature, the combination of weights and time constants of the relaxation processes contributing to the overall transient results in the “amplitude” of the normalized transient, i.e., (1-G(1800s)/G(5s)). Whereas a higher amplitude was observed for 3CCH than for 3CCO, similar amplitudes were observed for the chemically crosslinked scaffolds.

### 3.3. Cells Studies

Figure 6 clearly shows that the three-dimensional scaffolds are not cytotoxic, reflecting a progressive growth of the number of cells (in addition, it can be corroborated in Figure 7 and Figure 8 for up to 72 h, as extensively reported at literature [36,65,66]). This progressive growth is more significant in the case of materials crosslinked with EDC/NHS than those with glutaraldehyde, probably due to the toxicity of the latter. However, it is encouraging that the cells survive in this medium because, as previously discussed, the rheological properties are better when they are crosslinked with glutaraldehyde instead of carbodiimide.

The typical features of these three-dimensional scaffolds are interconnected pores between 100 and 150 μm in diameter (seen in the Figure 7), and areas that have junctions through wire-shaped structures due to collagen fibers, as seen in the SEM micrographs [33,61,87,88]. The change in the surface over time happens for two reasons, the interaction of the biological media with the surface and the simultaneous insertion of the cells through the interconnected pores.

In the first place, the reaction with glutaraldehyde is simple from the point of view of its chemical mechanism (just one reaction for the imine formation, Figure 2a), so although it changes a little in superficial aspects, the change is not as substantial as with the carbodiimide system. Most notable is the loss of boundaries between the layers—better defined in the case of non-crosslinked materials (Figure 2a,c and Figure 7) [31,89].

In the case of EDC/NHS, the chemical reactions formed by complex mechanisms based on the production, generation, and regeneration of stable and unstable intermediaries until the formation of the peptide bonds is achieved, may cause the pores covering the surface of the crosslinked scaffold. In addition, the C–N peptide bond has a shorter bond distance than the classical C–N bonds of chemical structures, due to the electronic effects around the alpha carbon, which makes the peptide bond in physiological media without enzymatic alterations a stable structure with some strength (bond energy ≈ 300 kJ/mol; bond distance ≈ 0.15 nm) [90,91].

On the other hand, cell adhesion can be clearly visualized by increasing the size of the white dots in the form of spheres on the surfaces of the three-dimensional scaffolds, perfectly distinguishable from the calcium phosphate filler, whose shape is not spherical. It can be observed in the second and fourth columns of Figure 7 that as time passes, the cells proliferate. The samples crosslinked with the EDC/NHS system were less toxic and were easily removable from the scaffold structures with water washing. In addition, the hydroxyapatite’s (but other calcium phosphates too) aggregate size was below 160 µm, but the humid synthesis method led to obtaining normal particle groupings with different irregular shapes [40,44,70], while the chondrocytes had 10–20 µm diameters and almost perfect spherical shapes [60,92].

In Figure 8A,E,I,M, numerous green dots corresponding to living cells within the three-dimensional structure are clearly observable, as is their proliferation from 3 to 14 days. In contrast, in Figure 8D,H,L,P the cells were meant to appear red if there was cellular apoptosis. That there was no relevant apoptosis according to the results obtained. In the rest of Figure 8, the structure of the scaffold can be seen, alongside the many nuclei of cells stained blue with DAPI, which relate to living cells, indicating a high survival rate.

It is important to note that only the results obtained for the scaffold crosslinked with EDC/NHS were shown, because glutaraldehyde could cause confusion when evaluating the results due to the crosslinking agent having self-fluorescence at the working wavelengths of this technique [93,94]. It was shown, however, in Figure 7 that the cells also adhere to and proliferate in the three-dimensional structure when crosslinked with glutaraldehyde, despite the recognized toxicity of the crosslinker.

## 4. Conclusions

In this study, novel 3D tri-layered scaffolds intended for the restoration of cartilaginous tissue were obtained. They consisted of three layers with a concentration gradient: the bottom (bone) layer was based on collagen, chitosan, and calcium phosphates to improve bone adhesion; the intermediate layer consisted of the mixture of collagen and chitosan or ELR; and finally, the top layer was based again on collagen and chitosan with a different ratio than that of the middle layer. Both physical and chemical crosslinking were used.

The scaffolds were characterized by the combination of several experimental techniques: morphological characterization (SEM micrographs), chemical-physical characterization (FTIR, XRD, and swelling behavior), and the characterization of mechanical properties through rheological measurements. SEM images showed a highly porous and interconnected structure with potential bioactivity in its bone layer due to the high concentration of Ca-P. The chemical crosslinking process significantly increases the rheological properties, while maintaining adequate matrix-fluid absorption and interaction. By following the BPVE model, it can be concluded from the analysis of the rheological data that two different physical mechanisms (poroelasticity and intrinsic viscoelasticity) dominate in different frequencies and time scales.

The rheological measurements showed mechanical properties similar to those required by natural tissues and to those in the literature—in some cases, the values we attained were better than those from the literature. The biological behavior of the material showed non-toxicity through a progressive increase in the population of exposed cells, which together with its rheological behavior makes it a promising candidate to be used as a matrix in cartilage tissue engineering.

## Figures and Tables

**Figure 1 polymers-13-00907-f001:**
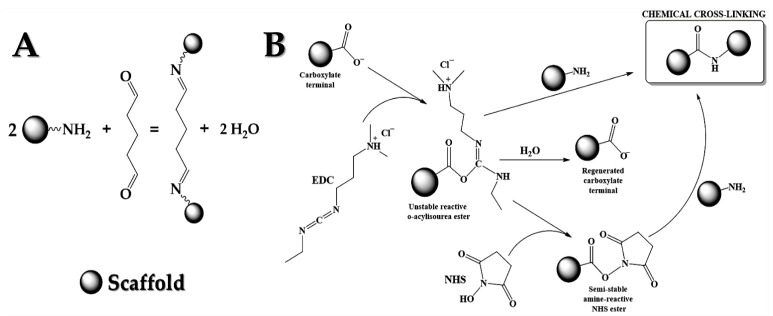
Representation of the crosslinking mechanism for the scaffold: (**A**) with glutaraldehyde for the formation of imine compounds as a result of crosslinking process; (**B**) with the EDC/NHS system for the formation of an amide bond, similar to amino acids in structure but more labile to interaction with the physiological medium.

**Figure 2 polymers-13-00907-f002:**
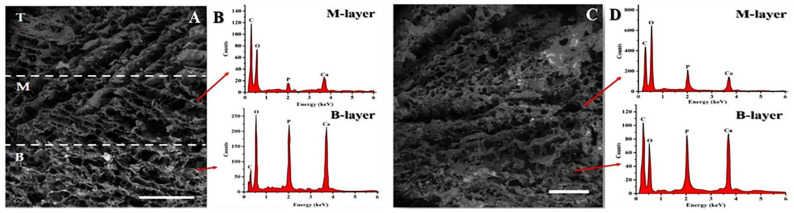
SEM (**A**) and EDX (**B**) of the 3CCO scaffold divided into (B)ottom, (M)edium and (T)op layers. SEM (**C**) and EDX (**D**) of the 3CCO.G scaffold. The frontiers between layers (A) are easy to identify by the changes in the shades of gray. Bar = 1 mm.

**Figure 3 polymers-13-00907-f003:**
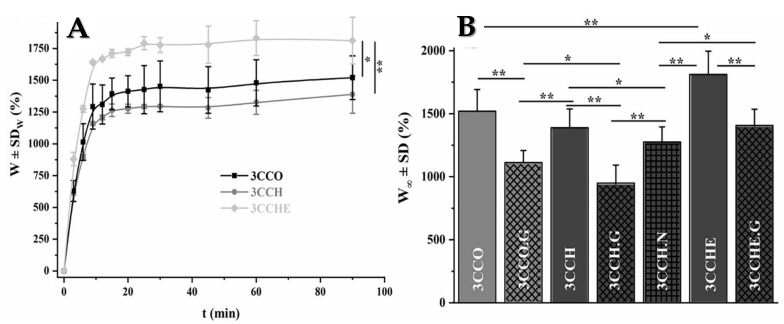
(**A**) Swelling profiles of the non-crosslinking samples and (**B**) maximum swelling after immersion for 90 min of all the scaffolds. Light gray indicates the use of OCP as the Ca-P material, and dark gray, the use of HAP. Non-filled bars denote non-crosslinking samples, while diagonal crosses denote the use of glutaraldehyde; straight crosses denote the use of the EDC/NHS crosslinking system.

**Figure 4 polymers-13-00907-f004:**
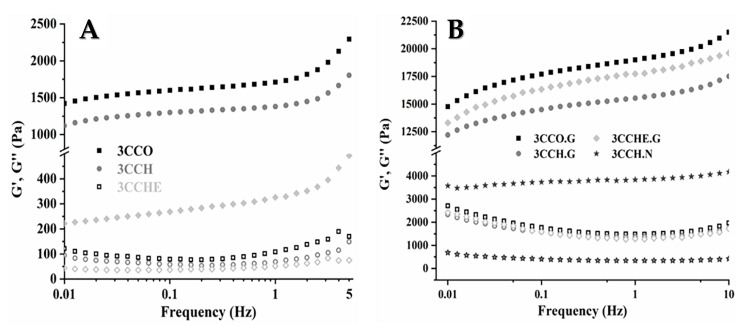
G’ (fill) and G″ (open) moduli of physically (**A**) and chemically (**B**) crosslinked scaffolds. Each curve corresponds to the average of three different measured samples. Error bars have been omitted for clarity.

**Figure 5 polymers-13-00907-f005:**
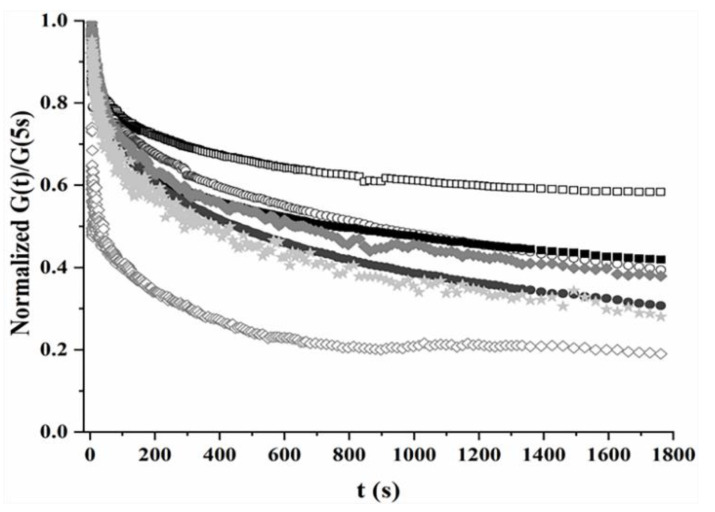
Normalized relaxation modulus transients for physically (open) and chemically (fill) crosslinked scaffolds. (☐) 3CCO, (◯) 3CCH, (◇) 3CCHE, (■) 3CCO.G, (●) 3CCH.G, (◆) 3CCHE.G, (🟊) 3CCH.N. Each curve corresponds to the average of three different samples measured. Error bars have been omitted for clarity.

**Figure 6 polymers-13-00907-f006:**
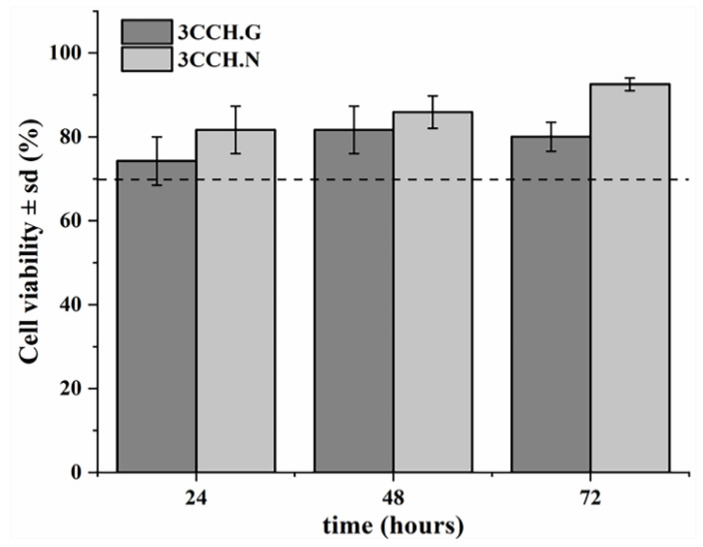
Cell viability assay results of crosslinking materials 3CCH.G and 3CCH.N at 24, 48, and 72 h, revealing that the scaffold is non-cytotoxic. 3CCH.N at 72 h was the only instance with significant differences from all the rest (*p* < 0.001).

**Figure 7 polymers-13-00907-f007:**
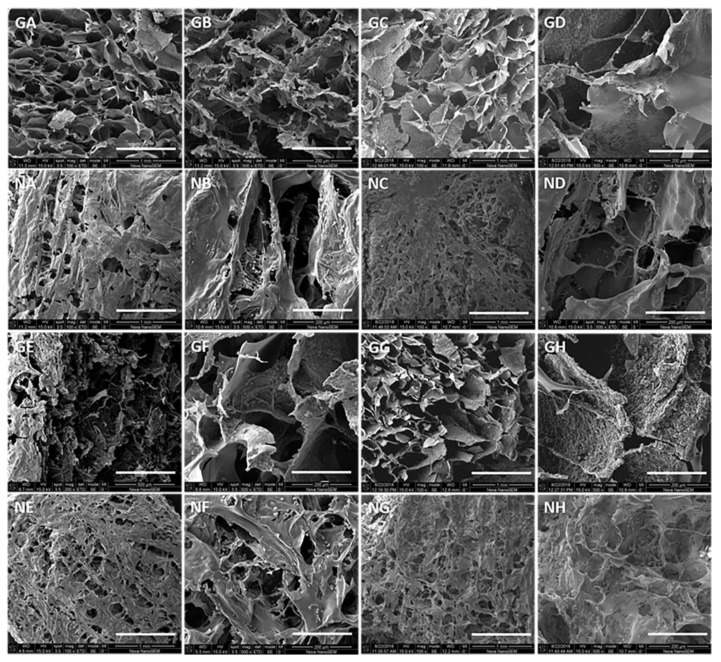
SEM of crosslinked scaffolds. The first letter in the code means crosslinked with (G)lutaraldehyde or EDC/(**N**)HS. For the second letter in the code: (**A**) or (**B**) = 3 days; (**C**) or (**D**) = 7 days; (**E**) or **F** = 10 days; and (**G**) or (**H**) = 14 days. For the first and third columns, bar = 1.

**Figure 8 polymers-13-00907-f008:**
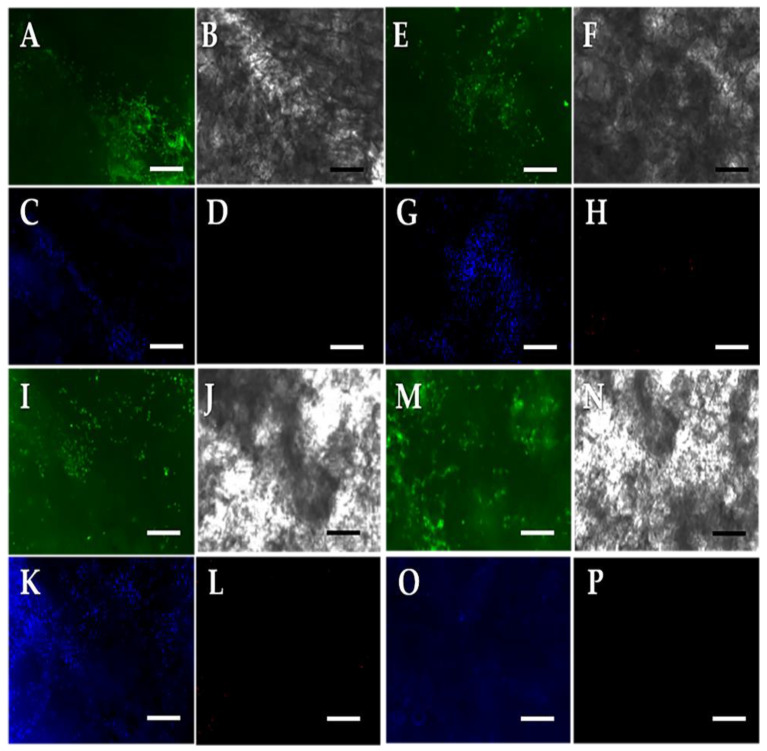
Confocal micrographs of a representative slice of scaffold crosslinked with EDC/NHS into which chondrocytes had been seeded on the top of surface. (**A**) Living cells, which appear in green, (**B**) scaffold structure, (**C**) cells stained with DAPI, and (**D**) dead cells, which appear in red—three days in. The letter sequences have the same significance as ABCD but changed with evaluation time: (**E**–**H**) for 7 days, (**I**–**L**) for 10 days, and (**M**–**P**) for 14 days. Micrographs with glutaraldehyde not shown due to interference of the structure. Bar = 200 µm.

**Table 1 polymers-13-00907-t001:** Sample codes and compositions of the scaffolds. **3CC** means 3 layers with collagen and chitosan. B, M, and T layers mean bottom, medium, and top layers, and bold letters define the specific composition in each scaffold. O—OCP, H—HAP, E—ELR, G—glutaraldehyde; N—NHS/EDC.

Sample Code	B-Layer	M-Layer	T-Layer	Cross-Linked
3CC**O**	COL:CHI (1:1) + 2% Ca-P	COL:CHI (1:1)	COL:CHI (3:1)	No
3CC**H**		COL:CHI (1:1)		No
3CC**HE**		COL:**E**LR(1:1)		No
3CC**O.G**		COL:CHI (1:1)		Yes (**G**)
3CC**H.G**		COL:CHI (1:1)		Yes (**G**)
3CC**H.N**		COL:CHI (1:1)		Yes (**N**)
3CC**HE.G**		COL:**E**LR(1:1)		Yes (**G**)

**Table 3 polymers-13-00907-t003:** Slope values of the linear region of Appendix A obtained by linear regression of the experimental data.

Sample	Slope (Pa/Hz½)
3CCO	166 ± 3
3CCH	115 ± 3
3CCHE	71 ± 2
3CCO.G	1010 ± 20
3CCH.G	850 ± 20
3CCH.N	139 ± 7
3CCHE.G	920 ± 20

**Table 4 polymers-13-00907-t004:** Fitting parameters of the experimental data of Figure 5 to Equation (2) (mean ± SE). G% = (G(1800s)/G(5s)) × 100.

Parameter	3CCO	3CCH	3CCHE	3CCO.G	3CCH.G	3CCH.N	3CCHE.G
G_eq_	0.576 ± 0.006	0.29 ± 0.06	0.208 ± 0.004	0.387 ± 0.003	0.267 ± 0.003	0.27 ± 0.01	0.31 ± 0.03
G_1_	0.06 ± 0.02	0.17 ± 0.02	3 ± 1	0.247 ± 0.003	0.155 ± 0.002	0.12 ± 0.03	0.212 ± 0.007
$\tau_{1}$(s)	8 ± 4	7 ± 1	1.9 ± 0.3	8.4 ± 0.2	12.3 ± 0.4	8 ± 3	23 ± 1
G_2_	0.10 ± 0.01	0.14 ± 0.02	0.334 ± 0.005	0.192 ± 0.002	0.218 ± 0.002	0.20 ± 0.03	0.18 ± 0.02
$\tau_{2}$(s)	50 ± 10	150 ± 30	210 ± 10	77 ± 2	85 ± 2	40 ± 10	180 ± 30
G_3_	0.207 ± 007	0.41 ± 0.03	---	0.272 ± 0.002	0.406 ± 0.002	0.39 ± 0.01	0.316 ± 0.009
$\tau_{3}$(s)	530 ± 60	1400 ± 400	---	860 ± 30	820 ± 20	460 ± 50	1200 ± 300
G%	58%	39%	19%	42%	31%	31%	37%
$R_{fit}^{2}\;$(%)	98.23	99.07	95.89	99.97	99.98	98.79	99.89

## Data Availability

Not applicable.

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
