# Peer review of "The Effects of Crosslinking on the Rheology and Cellular Behavior of Polymer-Based 3D-Multilayered Scaffolds for Restoring Articular Cartilage"

_polymers, 2021, doi:10.3390/polym13060907_

Round 1

Reviewer 1 Report

The purpose of this study was to investigate the effect of cross-linking methods on the rheology and cellular behavior of polymer-based 3D-multilayered scaffolds developed for cartilage regeneration. The study is well designed, and the wide array of methods used to analyze the research question allow a comprehensive insight into the topic. The results will add new knowledge in this research field. Figures and supplemental figures are adequate.

However, there are some aspects which need to be corrected to improve the quality of the manuscript:

  1. Double-check English grammar;
  2. Introduction can be improved with the inclusion of additional data (P2L51) from a previous study showing that stem cells combined with a natural scaffold based on platelet-rich plasma can regenerate, in an organotypic model, both soft and hard tissues https://doi.org/10.1007/s00784-021-03840-9;
  3. P3L98- reference 16 is not appropriately introduced in the document;
  4. Introduction must end with the formulation of a null hypothesis to be tested in this study;
  5. Materials and methods numbering are incorrect, must start with 2, and subsections numbering also needs to be double-checked;
  6. Table 1 needs additional information to be self-explanatory (what means B, M and T-layers need to be explained in the legend, also abbreviatures used in the table should be identified in the legend);
  7. Figure 1 will benefit from full identification of abbreviatures;
  8. P7L249-251 this sentence is not part of materials and methods section and must be moved to discussion;
  9. Results and discussion need re-numbering;
  10. Figures 7 and 8 would allow a better reading experience if they are presented in bigger format.

Reviewer 2 Report

I have reviewed a manuscript “The effect of crosslinking on the rheology and cellular behavior of polymer-based 3D-multilayered scaffolds for restoring articular cartilage”. It is a very nice study with interesting results. I think it is suitable for the publication after addressing the following comments:

Comment 1: please give more details of the experiments especially the instruments (model, company, country). For example, there is no information regarding the model of confocal microscopy used in this study.

Comment 2: please add a schematic illustration at the end introduction section that describes the overall view of the study (aim, fabrication process)

Comment 3: since crosslinking method significantly changes the mechanical properties of the scaffold, I would suggest explaining more about the chemical and ionic crosslinking method. Following references might be helpful:

CHEMICAL:

https://pubs.acs.org/doi/full/10.1021/acsabm.0c00169

https://onlinelibrary.wiley.com/doi/full/10.1002/pat.4654

https://www.future-science.com/doi/10.2144/btn-2018-0083

IONIC:

https://pubs.rsc.org/fi/content/articlehtml/2020/nr/d0nr02581j

https://pubs.rsc.org/en/content/articlehtml/2020/tb/d0tb00627k

Comment 4: Please use the same labeling for all figures. For example in figure 1, you have used capital letters for the labeling while other figures are small letters.

Comment 5: How did you measure and calculate the cell viability?

Round 2

Reviewer 2 Report

Respected authors have addressed the comments very well. I think it is suitable for publication as it is. For comments 3, I think Fig A is more appealing.